# Evaluation of Physicochemical Characteristics and Sensory Properties of Cold Brew Coffees Prepared Using Ultrahigh Pressure under Different Extraction Conditions

**DOI:** 10.3390/foods12203857

**Published:** 2023-10-21

**Authors:** Shiyu Chen, Ying Xiao, Wenxiao Tang, Feng Jiang, Jing Zhu, Yiming Zhou, Lin Ye

**Affiliations:** 1School of Perfume and Aroma Technology, Shanghai Institute of Technology, Shanghai 201418, Chinazhouymsit@126.com (Y.Z.); 2School of Food and Tourism, Shanghai Urban Construction Vocational College, Shanghai 201415, China; 3Coffee Professional Committee, Shanghai Technician Association, Shanghai 200050, China; 4Shanghai Acme Academic School, Shanghai 200062, China

**Keywords:** cold brew, ultrahigh pressure, physicochemical characteristics, sensory evaluation

## Abstract

Although cold brew coffee is becoming increasingly popular among consumers, the long coffee extraction time is not conducive to the further development of the market. This study explored the feasibility of ultrahigh pressure (UHP) to shorten the time required for preparing cold brew coffee. The effects of pressure and holding time on the physicochemical characteristics and sensory evaluation of UHP-assisted cold brew coffee were also determined. The extraction yield; total dissolved solid, total phenol, and melanoid content; antioxidant capacity; and trigonelline and chlorogenic acid contents of UHP-assisted cold brew coffee increased as the pressure increased. The extraction yield and the total dissolved solid, total phenol, total sugar, and chlorogenic acid and trigonelline contents were higher when the holding time was longer. The HS-SPME-GC/MS analysis demonstrated that the furan, aldehyde, and pyrazine contents in coffee increased as the pressure and holding time increased. The pressure did not significantly impact the concentrations of volatile components of esters and ketones in coffee samples. However, the increase in holding time significantly increased the ester and ketone contents. The sensory evaluation results revealed that as pressure rose, the intensities of nutty, fruity, floral, caramel, and sourness flavors increased, whereas bitterness and sweetness decreased. Longer holding time increased nutty, caramel, sour, bitter, sweet, and aftertaste flavors. Principal component analysis (PCA) results indicated that holding time is a more crucial factor affecting the physiochemical indices and flavor characteristics of coffee. UHP can shorten the preparation time of cold brew coffee. Pressure and holding time significantly affected the physiochemical indices and volatile components of UHP-assisted cold brew coffee. UHP-assisted cold brew coffee had lower bitterness, higher sweetness, and a softer taste than conventional cold brew coffee.

## 1. Introduction

Coffee is among the three most consumed beverages in the world. It is one of the biggest market segments for drinks and is expected to reach global revenues of USD 585 billion at home and abroad by 2025, largely driven by a continued boom in specialty coffees in the food service industry [1]. Consumers are recently in the pursuit of higher quality coffee, and so the demand for cold brew coffee is gradually increasing. Cold brew coffee is prepared by cold brewing. In cold brewing, coffee beans are soaked in cold water (5 °C) for at least 12 h to extract slight flavor substances such as floral and fruity aromas as well as a coffee liquid with a softer and slightly sweet taste [2]. Cold brew coffee is often described as sweeter or less acidic than hot brew coffee [3]. However, production of cold brew coffee is time-consuming because of its long extraction time (approximately 12–24 h).

Current research on cold brew coffee focuses on its chemical composition and sensory properties. According to some studies, the content of volatile and nonvolatile compounds in cold brew coffee depends on time–temperature roasting conditions [4]. Particle size, extraction time, and coffee type affect the physicochemical and sensory properties of cold brew coffee, leading to variations in flavor profiles [5]. The degree of roasting affects chlorogenic acid, trigonelline, and other compounds in coffee. Rao and Fuller [6] reported that deeper roasting decreases the concentrations of compounds in both hot and cold brew coffee, whereas the total antioxidant capacity is sensitive to roasting degree only in cold brew coffee.

Numerous studies investigated the physiochemical indices of cold brew coffee. However, reports on reducing the time of cold brew coffee extraction are limited. Caudill et al. [7] proposed that a brief period of heat treatment before cold brewing can accelerate cold brew coffee extraction while reducing production costs. According to Ahmed et al. [8], ultrasonication and agitation of cold brew coffee considerably influences its physicochemical properties. These unconventional methods are beneficial for nutrient extraction from cold brew coffee, including for obtaining high antioxidant phenolic substances. Vacuum cycling can significantly accelerate cold brew coffee extraction, and the extraction rate is the maximum at 65 min [9]. Although these methods can reduce the time for extracting cold brew coffee, their extraction time is still up to 1 h.

Therefore, other methods need to be explored. This study investigates a more efficient method for shortening the extraction time of cold brew coffee. Recently, ultrahigh pressure (UHP) technology has been extensively used in food processing because it has little effect on the nutritional value, sensory quality, and texture of food [10]. Ma et al. [11] reported that UHP treatment can modify the protein structure of steamed oats, make their surface more porous and uneven, increase the water absorption rate, reduce the hardness of the steamed oats, and serve as a positive player in improving the edible quality of the steamed oats. The UHP method was used to extract chlorogenic acid (CGA) and cynaroside from L. japonica flower buds, and a greater extent of tissue structure rupture was observed in the UHP-treated samples [12]. Xi et al. [13] highlighted that UHP processing of green tea extract greatly influenced the total phenolic content and antioxidant activity. This is because this technology destroyed the structure of tea tissues, cell walls, membranes, and organelles (especially vacuoles), thereby enhancing the mass transfer of the solvent to leaf materials and soluble components to solvents. The UHP method has the advantages of a short extraction time, high extraction rate, and low energy consumption over other extraction methods [12]. Therefore, it is speculated that UHP technology can improve the extraction efficiency by destroying the cell structure, thus shortening the preparation time of cold brew coffee. Only Zhang et al. [14] reported the application of the UHP method to pretreat whole coffee beans and soak them for 12 h and investigated its effect on physicochemical properties. However, the direct use of the UHP method for extracting coffee powder to produce cold brew coffee, the flavor characteristics of UHP-assisted cold brew coffee, and the ability of UHP to accelerate cold brew extraction remain unexplored.

This study explored the impact of the application of the UHP method on the acceleration of cold brew coffee extraction. It also assessed the effects of pressure and holding time on various physiochemical parameters, nonvolatile components, sensory evaluation, and volatile components of UHP-assisted cold brew coffee.

## 2. Materials and Methods

### 2.1. Reagents and Chemicals

Caffeine (99%), CGAs (99%), trigonelline (99%), and 1,1-diphenyl-2-picrylhydrazyl radical (DPPH) were purchased from Yuanye Biotechnology Co., Ltd. (Shanghai, China). Titan Technology Co., Ltd. (Shanghai, China) provided 2-octanol and 2,2’-azinobis-(3-ethylbenzthiazoline-6-sulphonate) (ABTS). Sigma–Aldrich Chemical Co., Ltd. (Shanghai, China) provided Folin-phenol (2M).

### 2.2. Coffee Sample Preparation

Arabica coffee beans cultivated in Ethiopia were purchased from Yanbei Coffee Co., Ltd. (Shanghai, China). Coffee samples were roasted using a drum roaster (SR5 Manual Coffee Roaster, Piła, Poland) for a roasting time of 11.5 ± 0.5 min and at a final temperature of 205.0 ± 1.0 °C until a medium roast level (L value: 24–26) was attained. The coffee beans were ground using a commercial grinder (EK43s, Mahlkonig, Portland, Italy) until the grain size was 400–600 μm. UHP-assisted cold brew coffee was prepared with 15 g coffee powder and 210 g water (5 °C) and then weighed in a bottle. The bottle was immediately placed into a UHP unit (UHP-600 ultrahigh voltage equipment, Baotou Kefa High Voltage Technology Co., LTD., Baotou, China) and treated for 10–30 min at 300–500 MPa. When the pressure reached the atmospheric level, the coffee samples were immediately filtered through a filter paper and stored in a 5 °C refrigerator. While preparing conventional cold brew coffee, the same powder-to-water ratio was mixed and soaked in a 5 °C refrigerator for 12 h, strained using a coffee filter, and stored in a 5 °C refrigerator.

### 2.3. Extraction Yield and Total Dissolved Solid

The total dissolved solid (TDS) value of coffee was measured using a TDS refractometer (Pal-Coffee, ATAGO, Minato, Japan) [15]. The extraction yield (EY) reflects the relationship between the total extract weight obtained (Wb), the weight of ground coffee used in extraction (Wgc), and TDS and is defined as follows: EY (%) = (TDS × Wb/Wgc) × 100 [16].

### 2.4. Total Phenol Compounds, Total Sugars, and Total Titratable Acidity

To achieve total titratable acidity (TTA), 50 mL of the coffee extract was titrated with 0.1 M NaOH until pH 6.5 was attained [6]. The total sugar (TS) concentration in the coffee brew was determined using the phenol-sulfuric acid method [17]. Total phenol compounds (TPC) in coffee liquid were determined using the BILGE method [18]. To evaluate TPC, 0.5 mL coffee liquid was diluted 50 times. Then, 0.25 mol/L Folin-phenol reagent was added to the coffee liquid, mixed, and allowed to stand for 3 min. Subsequently, 1 mL 15% Na_2_CO_3_ was added to this mixture, mixed, and centrifuged at 120 rpm and 25 °C for 1 h away from light. An ultraviolet spectrophotometer was used to measure absorbance at a wavelength of 765 nm. The pyrogallic acid standard curve was used to calculate TPC.

### 2.5. Antioxidant Capacity and Melanoidins

The coffee extract was diluted at a 1:19 ratio, and its light absorption value was determined at 420 nm [19]. The melanoid content was estimated at a light absorption value of 1.1289 L·g^−1^·cm^−1^.

The scavenging capacities of ABTS and DPPH were determined to evaluate the antioxidant activities of the coffee brew. DPPH was determined using the method of Dong et al. [20], with some modifications. ABTS was determined using the method of Gorecki et al. [21]. The absorbance values were expressed as mmol/L trolox with trolox solution (10–100 μmol/L).

### 2.6. CGAs, Trigonelline, and Caffeine

Following the method of Córdoba et al. [4] with slight modifications, CGA, trigonelline, and caffeine components of coffee brews were measured through high-performance liquid chromatography (HPLC). An LC-20A HPLC (Shimadzu, Nishinokyo, Japan) with a photodiode array detector was used for quantitative analysis. Caffeine was separated using a WondaSilTM C-18 column (150 mm × 4.6 mm × 5 µm, Shimadzu, Japan) at 30 °C. The mobile phase included 24% methanol and 76% water. Trigonelline was separated using a WondaCract ODS-2 column (150 mm × 4.6 mm × 5 µm, Shimadzu, Japan) at 30 °C. The mobile phase included 12% methanol and 88% water. CGAs were separated using the WondaCract ODS-2 column (150 mm × 4.6 mm × 5 µm, Shimadzu, Japan) at 30 °C. The mobile phase included acetonitrile and 1% acetic acid (ratio is 15:85 (*v*/*v*)). The injection volume was 10.0 µL, and the mobile phase was used at a 1.0 mL/min flow rate. CGAs and trigonelline were measured at 260 nm, and caffeine was measured at 272 nm. The concentrations of caffeine, trigonelline, and CGAs were calculated based on a regression equation of their concentrations as HPLC standard references.

### 2.7. Volatile Compounds

Volatile compounds in the coffee brew were quantified using the optimized Yu [22] method with minor revisions. Headspace solid-phase microextraction (HS-SPME) and gas chromatography/mass spectrometry (GC–MS) were performed to analyze these volatile compounds. The volatiles of vials in the headspace were extracted using a divinylbenzene/carboxen/polydimethylsiloxane SPME fiber of 50/30 µm film thickness with a manual SPME holder (Supelco, Bellefonte, PA, USA). Then, 5 mL of sample and 20 μL of the internal standard 2-octanol were added to the GC vial. In a 60 °C water bath, the sample was equilibrated for 15 min and absorbed for 30 min, followed by 2 min desorption in the GC injector at 200 °C in a splitless mode. After desorption was completed, the compounds were further separated through GC–MS. A gas chromatograph coupled with a mass spectrometer (Shimadzu, TQ-80, Japan) was equipped with the capillary RTX-WAX column (Shimadzu, 30 m × 0.25 mm × 0.25 μm film thickness, Japan). The column oven was programmed from 40 °C (after 2 min) to 130 °C at a rate of 2/min, increased to 220 °C at a rate of 4 °C/min, further increased to 250 °C at 10 °C/min, and held at the final temperature for 5 min. Helium was selected as a carrier gas and incorporated at a 1.6 mL/min flow rate, and energy voltage was maintained at 70 eV. The peak area ratio of each compound was calculated according to the peak area ratio of the internal standard.

### 2.8. Sensory Evaluation

Following the methodology of the Specialty Coffee Association (SCA), a trained team of 6 members, who were certified as Arabica Q-Grader by the Coffee Quality Institute (CQI), conducted the sensory evaluation in the sensory analysis room. Two sessions (2 h each) were conducted to familiarize the panel with the sensory vocabulary selected, their definitions, and their intensities (World Coffee Research, 2017). All referees had extensive experience and were trained in the cupping process. They provided informed consent for the attributes of the test sample. All samples were served at room temperature (20 °C ± 3 °C) and randomly evaluated in triplicate. A trained member sipped each sample and rated the odor intensity perceived retronasally. Then, the members could take a second sip to rate the taste and mouthfeel attributes. Before sipping different samples, they had to gargle with warm water to minimize the carry-over effect. A list of specific descriptors and constants was selected and used for the sensory analysis. Flavor attributes included were as follows: floral, fruity, nutty, spices, caramel, astringency, flavor, sweetness, sourness, bitterness, body, aftertaste, and overall intensity. The intensity of these flavor attributes was determined using a 0- to 15-point scale with 0.5 increments (0 = none; 15 = extremely intense) [23].

### 2.9. Statistical Analysis

The experiments were triplicated, and the analyses were duplicated (3 × 2). Data obtained were expressed by the average value of standard deviations. SPSS 23.0 software was used for statistical analysis of the data, and significance was defined as *p* < 0.05. GraphPad Prism 9.0 and SIMCA 14 software were used for constructing column graphs and performing principal component analysis (PCA), respectively.

## 3. Results and Discussion

### 3.1. Physicochemical Properties under Different Extraction Conditions

Table 1 presents the physicochemical properties (TDS, EY, TTA, TPC, TS, DPPH, ABTS, and melanoidins) of the coffee samples. As the pressure and holding time increased, TDS increased from 1.11% to 1.22% and 1.11% to 1.24%, as excessive pressure destroys the cellular matrix, which makes the extraction of compounds easier [24]. Cold brew coffee at 0.1 MPa and 12 h was used as the control. The samples of UHP-assisted cold brew coffee had higher TDS and EY when the pressure was >300 MPa and time was >20 min. Zhai et al. [25] exhibited a similar phenomenon during ultrasound-assisted coffee extraction. They revealed that more water can move into the cells because the tissues and cell walls are disrupted through sonoporation during ultrasonication, which leads to increased cell membrane permeability and passing of more soluble solids through the cell membrane and compensates for the time effect on cold brew. Thus, the UHP method can achieve TDS and EY similar to conventional cold brew coffee within 20 min. This proves the feasibility of the UHP method for reducing the excessively long time of cold brew coffee extraction.

As the pressure and holding time increased, the TS and TPC values of UHP-assisted cold brew coffee increased significantly (*p* < 0.05). However, the TA value exhibited no significant change (*p* > 0.05). The increase in pressure and holding time increased the temperature of the enclosed space, thereby accelerating the release of sugars and phenols [26]. Fuller and Rao [27] reported that at the beginning, CGA was released rapidly, and over time, that is, by 400 min, CGA continued to exhibit an increasing trend. This indicated that phenolic substances in coffee are more significantly released under the influence of holding time than under the influence of pressure. Similar phenomena were reported by other studies investigating the effect of UHP treatment of tea leaves in water. They observed that pressure can enhance TPC extraction [28]. Coffee is composed of several low-molecular-mass compounds such as citric acid, malic acid, quinic acid, succinic acid, and gluconic acid. Because of their high water solubility, these acidic components can more easily dissolve in the early extraction stages, and hence, no significant difference is observed in TA extraction [29].

The antioxidant activity of UHP-assisted cold brew coffee was determined using the DPPH and ABTS free radical scavenging methods. The scavenging abilities of DPPH and ABTS free radicals increased as the pressure and retention time increased. CGAs are the main phenolic compounds in coffee beverages [30]. Compared with the other bioactive compounds investigated, CGA may exert a more significant impact on the antioxidant capacity of coffee. However, as the pressure and holding time increased, the melanoid content also increased, and the trend of antioxidant capacity of coffee was the same as that of the melanoid content. Melanoid is reported to help maintain high antioxidant levels in coffee [31]. According to Vignoli et al. [32], the UHP method can cause significant structural changes in coffee powder, thereby enhancing the extraction efficiency of antioxidant compounds. Consequently, the DPPH and ABTS free radical scavenging abilities of UHP-assisted cold brew coffee can surpass that of cold brew coffee at 500 MPa and 30 min, possibly due to the increase in melanoid content.

Caffeine is a heat-stable substance, and so higher water temperature causes no increase in caffeine extraction [33]. In fact, the extraction of UHP-assisted cold brew coffee saturates the pores within and between the particles, thereby promoting the rapid diffusion of caffeine through the solid matrix. Consequently, the caffeine concentration produced is almost identical to that produced by cold brew after 720 min. Therefore, Figure 1 shows that neither pressure and holding time nor the extraction method significantly influences caffeine extraction.

Figure 1B indicates that time has a greater impact on CGA content than pressure (*p* < 0.05). CGA is highly water-soluble and can be efficiently extracted at both low and high temperatures. Additionally, the CGA molecule is not restricted by intragranular pore diffusion processes. Higher CGA extraction can be achieved in a short time (10–30 min), but the amount extracted remains lower than that produced through cold extraction under atmospheric pressure. Complete CGA extraction requires soaking the coffee for a long period. Trigonelline is a relatively stable substance in cold brew coffee [34], but it may be easily affected by extraction conditions. Therefore, trigonelline content increases with pressure and holding time during the UHP-mediated extraction, which is quite different from conventional cold brew coffee.

### 3.2. Volatile Composition

Furan, pyrazine, ester, and aldehyde were the main volatile compounds found in UHP-assisted and conventional cold brew coffee (Table 2, Table 3 and Appendix A). Furthermore, minor volatile compounds such as alcohols, ketones, pyrroles, and phenols were also detected. Of note, (E)-beta-damascenone and furfural acetone were exclusively detected in conventional cold brew coffee. Pressure and holding time affected the total abundance of furans, aldehydes, and pyrazines, and these volatile components exhibited a higher abundance as pressure and holding time increased. The three volatile components of typical representative substances, namely furfural, 2-acetylfuran, 5-methylfurfural, and 2-ethylpyrazine, commonly produce nutty, caramel, and baking aromas [35,36]. The results of volatile components mostly matched the results of sensory evaluation. In the sensory evaluation results, a similar phenomenon was observed for the volatile components, and the sensory scores of nut and caramel also increased as the pressure and holding time increased. Although pressure showed no significant effect on the total volatile components of esters and ketones, especially furfuryl acetate, their contents increased significantly as the holding time increased. Specific substances, such as furfuryl acetate and linalool, are responsible for floral, fruity, and sweet flavors [37]. This may be one reason for the increased scores of floral, fruity, and sweetness flavors in the sensory evaluations.

Less significant changes were noted in the aroma of coffee samples under different pressures compared with at different times. For other substances with lower concentrations, as the pressure increased, the phenol content increased, the ether content decreased, and pyridine and pyrrole exhibited no significant difference (*p* > 0.05). Pyridine and pyrrole are associated with smoke, burnt, and other negative odor substances [36]. The amounts of volatile components of pyridine, pyrrole, phenols, and ethers also increased as the holding time increased. However, when the total amount of volatile components also increased, their relative proportions remained relatively stable and therefore may not cause negative sensory effects.

The furan content of UHP-assisted cold brew coffee was higher than that of conventional brew coffee only at 500 MPa and 30 min. Moreover, the aldehydes, esters, and ketones found in UHP-assisted cold brew coffee under 30 min were similar to those noted in conventional cold brew coffee. They are responsible for floral, fruity, and sweet tastes.

### 3.3. Sensory Analysis

Pressure significantly affected the sensory indicators of nutty, fruity, floral, caramel, sourness, and aftertaste flavors. As the pressure increased, the intensities of these indicators also increased, similar to the changes in the content of volatile components of pyrazines, furans, aldehydes, and esters detected through GC-MS (Table 4). The volatile components responsible for nutty, caramel, floral, and fruity flavors positively affected the sensory evaluation. Coffee samples with higher perceived acidity had higher CGA; however, the same trend was not observed for TTA. TTA is usually attributed primarily to CGAs [13]. It is also related to organic acids such as citric acid and malic acid. Bitterness, aftertaste, and astringency of coffee are closely linked to CGA, trigonelline, and caffeine concentrations [38]. Caffeine particularly strongly affects the bitterness of coffee. The holding time significantly affected trigonelline and CGA levels (*p* < 0.05), which explains why holding time more significantly affects the bitterness of UHP-assisted cold brew coffees.

Holding time had a similar effect on the sensory intensity as pressure. However, the change in sweetness with holding time was more significant than that with pressure (Table 5). This may be because the variation trend of sweet taste is similar to that of TS and other sweet volatile components such as furfuryl acetate, which are more affected by the retention time.

Conventional cold brew coffee had a higher sensory intensity than UHP-assisted cold brew coffee because of its longer extraction time and higher content of volatile components, which led to a more intense flavor profile. Compared with conventional cold brew coffee, UHP-assisted cold brew coffee exhibited higher sweetness and cleanliness levels and a softer taste, with lower levels of negative flavors such as bitterness, astringency, and vegetable flavors.

### 3.4. Principal Component Analysis

The PCA components represent explained variances of 51.1% and 16.3% for a 67.4% total variance explained (Figure 2). Based on the PC1 axis, the UHP coffee samples obtained under varying pressures and holding times were distinctly separated from the conventional cold brew samples. Samples extracted using the UHP method at a lower pressure (100–300 MPa) and shorter time (10–20 min) were located on the negative side of the PC1 axis, whereas those extracted at a higher pressure and longer holding time and conventional cold brew coffee samples were located on the positive side of the PC1 axis. T5 (500 MPa, 20 min) and CG (0.1 MPa, 12 h, 5 °C) were mainly associated with high TPC concentrations and high TTA, TDS, and antioxidant capacity. The acid, astringent, floral, and fruity flavors of these two samples were more pronounced. These samples consisted of ketones, aldehydes, alcohols, and esters as volatile components, with fruity and sweet flavors dominating the fragrances [39]. High-pressure and long-holding-time samples, such as T4 (300 MPa, 25 min), P4 (400 MPa, 20 min), and P5 (500 MPa, 20 min), exhibited the highest TS, melanoid, and caffeine concentrations and were located in the fourth quadrant. The volatile components in these samples were primarily furans, which resulted in a nutty aroma. UHP-assisted cold brew coffee samples produced at a lower pressure and shorter holding time were located in the second and third quadrants and had a lower physicochemical index, and they mostly had acetaldehyde, furfuryl methyl sulfide, ethanol, and methyl ethyl ketone as their volatile components, which contributed to other negative aromas. The increased pressure and holding time of high-pressure auxiliary cold brew coffee decreased the content of volatile ingredients, such as pyridine and pyrrole, in blemishes. The sample of UHP-assisted cold brew coffee under 30 min was regarded as the closest to the conventional cold brew coffee sample. They had similar physicochemical properties, volatile components, and sensory evaluation, represented by sourness, floral, and fruity flavors.

Interestingly, pressure and holding-time factors are located in the positive half of the PC1 axis, but holding time and pressure are located in the first and fourth quadrants, respectively. The PCA revealed that the holding-time factor was primarily responsible for CGAs, total phenols, aldehydes, alcohols, bitterness, fermented, astringency, spices, and body sensory notes. Moreover, the analysis revealed that the pressure factor was mainly associated with caffeine, melanoid, TS, and pyrazine contents. Therefore, the pressure and holding time maintained during extraction contributed to the extraction of volatile components and physicochemical indices of UHP-assisted cold brew coffee. The increased pressure was more conducive to the extraction of caffeine, melanoid, and TS and increased the extraction of nutty and sweet substances such as pyrazines and ketones in UHP-assisted cold brew coffee. The increased time was conducive to the extraction of CGA and TPC and increased the extraction of aldehydes, alcohols, and pyrroles in UHP-assisted cold brew coffee, contributing to other caramel, aromatic, and bitterness flavors. Moreover, holding time had a greater impact on the quality of UHP-assisted cold brew coffee than pressure.

## 4. Conclusions

This study reported the preparation of cold brew coffee using the UHP method. The physiochemical indices, nonvolatile components, volatile components, and sensory evaluation of coffee were investigated. UHP-assisted cold brew coffee has a shorter preparation time than conventional cold brew coffee. The prepared coffee has higher sweetness and cleanliness, lower bitterness, and a softer taste. Thus, UHP is a very effective treatment method. Pressure and holding time significantly affected physical and chemical indices and volatile components of UHP-assisted cold brew coffee, and holding time is a more crucial factor affecting physical and chemical indices and flavor characteristics of coffee.

Future research should be conducted to study the effects of other factors, such as the effect of roasting degree of coffee beans or water quality, on the UHP-assisted cold brew coffee extraction process. Meanwhile, the feasibility of using this new technology in commercial coffee chain stores requires further exploration. Finally, the mechanism and reasons underlying the acceleration of UHP-assisted cold brew coffee extraction should be confirmed in future.

## Figures and Tables

**Figure 1 foods-12-03857-f001:**
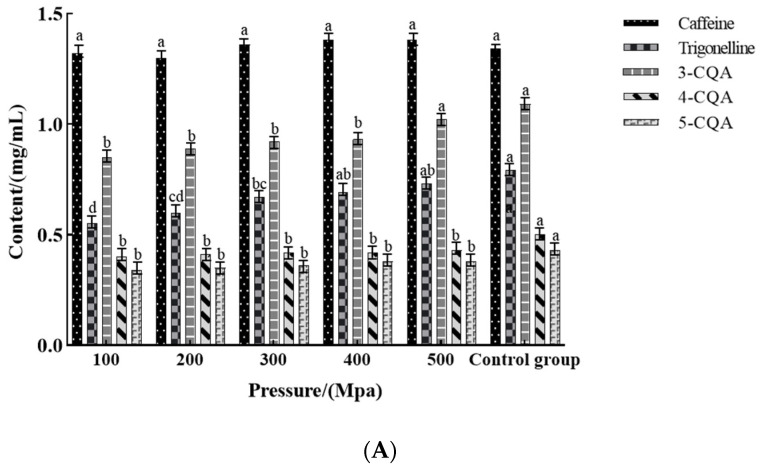
Influence of different pressures (**A**) and times (**B**) on the non-volatile components of UHP-assisted cold brew coffee. The superscript letters (a, b, c, and d) represent statistically significant differences between extraction conditions, as determined through one-way analysis of variance (*p* < 0.05). Control group: conventional cold brew (0.1 MPa, 12 h, 5 °C).

**Figure 2 foods-12-03857-f002:**
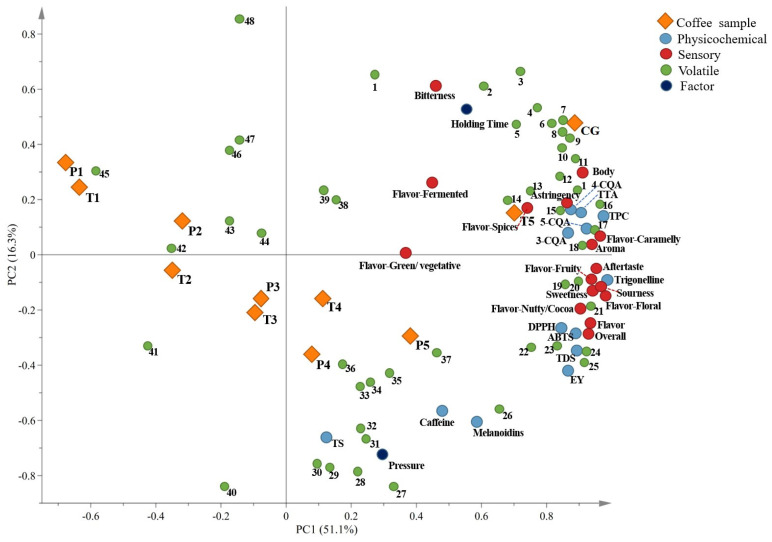
Principal component analysis diagram of cold brew coffee under different pressures and times. 1: 2-acetylpyrrole; 2: 2-formylpyrrole; 3: alpha-terpineol; 4: difurfuryl ether; 5: furfuryl propionate; 6: ethylcyclopentenone; 7: 5-methylfurfural; 8: 1-methyl-2-pyrrole formaldehyde; 9: 4-ethyl guaiacol; 10: 2-acetyl-1-methylpyrrole; 11: linalool; 12: furfuryl acetate; 13: 36, 6-theopyrazine; 14: o-cresol; 15: nerolol; 16: 2-ethyl-6-methylpyrazine; 17: 2-(furan-2-methyl-furan); 18: furfuryl alcohol; 19: furfural; 20: phenol; 21: 2-acetylfuran; 22: 2-methylbutyral; 23: 2, 5-dimethylpyrazine; 24: hazelnut pyrazine; 25: 2-ethylpyrazine; 26: isovalerate; 27: pyridine; 28: 2, 3-pentanedione; 29: 2, 3-butanedione; 30: 2, 3-hexadione; 31: 2-ethyl-1-hexanol; 32: furfuryl methyl ether; 33: 2, 3-dimethylpyrazine; 34: 2-propionylfuran; 35: 2, 5-dimethylfuran; 36: 3-ethylpyridine; 37: 37:2-acetyl-5-methylfuran; 38: 2-methylfuran; 39: 2, 6-dimethylpyrazine; 40: hexanol; 41: hexal (aldehyde C-6); 42: 3-hexanone; 43: methyl ethyl ketone; 44: 1-methylpyrrole; 45: furfuryl methyl sulfide; 46: 3, 4-hexanedione; 47: ethylene glycol diacetate; 48: benzaldehyde. P1: 100 MPa, 20 min; P2: 200 MPa, 20 min; P3: 300 MPa, 20 min; P4: 400 MPa, 20 min; P5: 500 MPa, 20 min. T1: 300 MPa, 10 min; T2: 300 MPa, 15 min; T3: 300 MPa, 20 min; T4: 300 MPa, 25 min; T5: 300 MPa, 30 min. CG: conventional cold brew (0.1 MPa, 12 h, 5 °C).

**Table 1 foods-12-03857-t001:** Effects of different extraction conditions on the physical and chemical indices of UHP-assisted cold brew coffee.

Extraction Conditions		Total Dissolved Solids/%	Extraction Yield/%	Titratable Acidity/(mL 0.1 mol/L NaOH)	Total Phenols Content/(mg/mL)	Total Sugar/(mg/mL)	DPPH/(Trolox/(mmol/L))	ABTS/(Trolox/(mmol/L))	Melanoidins/(mg/mL)
Pressure/(MPa) 20 min	100	1.11 ± 0.02 ^d^	15.74 ± 0.35 ^e^	0.31 ± 0.01 ^a^	3.14 ± 0.12 ^e^	0.80 ± 0.02 ^a^	5.19 ± 0.37 ^c^	2.34 ± 0.28 ^de^	4.68 ± 0.34 ^d^
	200	1.15 ± 0.03 ^cd^	16.82 ± 0.44 ^cd^	0.32 ± 0.01 ^a^	3.25 ± 0.13 ^d^	0.81 ± 0.03 ^a^	5.26 ± 0.35 ^c^	2.53 ± 0.30 ^d^	4.90 ± 0.27 ^c^
	300	1.18 ± 0.01 ^bc^	17.11 ± 0.32 ^bc^	0.32 ± 0.01 ^a^	3.40 ± 0.15 ^cd^	0.81 ± 0.02 ^a^	5.57 ± 0.46 ^bc^	2.64 ± 0.19 ^d^	5.23 ± 0.33 ^ab^
	400	1.22 ± 0.02 ^ab^	17.55 ± 0.45 ^b^	0.33 ± 0.02 ^a^	3.52 ± 0.14 ^cd^	0.82 ± 0.04 ^a^	5.86 ± 0.39 ^ab^	2.88 ± 0.24 ^ab^	5.35 ± 0.29 ^a^
	500	1.22 ± 0.03 ^ab^	17.88 ± 0.31 ^a^	0.32 ± 0.01 ^a^	3.66 ± 0.19 ^c^	0.86 ± 0.02 ^a^	5.94 ± 0.42 ^ab^	3.01 ± 0.33 ^a^	5.40 ± 0.41 ^a^
Holding time/(min) 300 MPa	10	1.11 ± 0.01 ^d^	16.10 ± 0.35 ^de^	0.30 ± 0.01 ^a^	3.01 ± 0.07 ^f^	0.75 ± 0.02 ^b^	4.65 ± 0.34 ^d^	2.32 ± 0.31 ^e^	5.16 ± 0.21 ^b^
	15	1.12 ± 0.02 ^d^	16.23 ± 0.30 ^de^	0.31 ± 0.01 ^a^	3.17 ± 0.13 ^d^	0.77 ± 0.03 ^ab^	4.92 ± 0.38 ^cd^	2.30 ± 0.29 ^e^	5.18 ± 0.32 ^b^
	20	1.18 ± 0.02 ^bc^	17.11 ± 0.29 ^bc^	0.32 ± 0.01 ^a^	3.40 ± 0.12 ^d^	0.81 ± 0.02 ^a^	5.57 ± 0.35 ^bc^	2.64 ± 0.26 ^d^	5.23 ± 0.30 ^ab^
	25	1.20 ± 0.01 ^ab^	17.58 ± 0.44 ^b^	0.32 ± 0.01 ^a^	3.71 ± 0.13 ^c^	0.81 ± 0.03 ^a^	5.58 ± 0.41 ^bc^	2.71 ± 0.23 ^d^	5.25 ± 0.27 ^ab^
	30	1.24 ± 0.02 ^a^	17.67 ± 0.47 ^ab^	0.34 ± 0.01 ^a^	4.07 ± 0.19 ^b^	0.83 ± 0.04 ^a^	6.05 ± 0.32 ^a^	2.94 ± 0.25 ^bc^	5.29 ± 0.38 ^ab^
Control group	0.1 MPa 12 h	1.21 ± 0.02 ^ab^	17.55 ± 0.39 ^b^	0.35 ± 0.02 ^a^	4.25 ± 0.22 ^a^	0.73 ± 0.03 ^b^	5.87 ± 0.45 ^ab^	2.89 ± 0.33 ^c^	5.17 ± 0.25 ^b^

Values are expressed as mean ± standard deviation. The superscript letters (a, b, c, d, e, and f) represent statistically significant differences between extraction conditions, as determined through one-way analysis of variance (*p* < 0.05). Control group: conventional cold brew (0.1 MPa, 12 h, 5 °C).

**Table 2 foods-12-03857-t002:** Volatile components of UHP-assisted cold brew coffee under different pressures.

Aroma Type	Content/(μg/L)					
	Pressure/(MPa)					Control Group
	100	200	300	400	500	
Furans	118.61 ± 6.72 ^d^	126.86 ± 7.05 ^cd^	130.93 ± 5.95 ^abc^	136.04 ± 5.47 ^b^	147.32 ± 6.84 ^a^	142.74 ± 8.43 ^ab^
Aldehydes	90.42 ± 4.92 ^c^	93.26 ± 6.43 ^bc^	94.30 ± 5.39 ^bc^	94.25 ± 4.92 ^bc^	101.48 ± 5.34 ^ab^	111.89 ± 6.53 ^a^
Esters	72.27 ± 4.33 ^b^	74.20 ± 4.50 ^b^	71.94 ± 5.13 ^b^	72.46 ± 4.85 ^b^	75.73 ± 5.31 ^b^	85.57 ± 6.45 ^a^
Pyrazines	40.78 ± 2.49 ^c^	43.47 ± 3.01 ^bc^	45.63 ± 2.88 ^b^	46.63 ± 2.97 ^b^	52.22 ± 3.85 ^a^	53.34 ± 4.13 ^a^
Alcohols	14.25 ± 0.95 ^d^	16.36 ± 0.73 ^c^	17.52 ± 0.98 ^bc^	17.44 ± 1.05 ^bc^	19.10 ± 1.47 ^b^	25.89 ± 1.59 ^a^
Ketones	15.53 ± 1.16 ^b^	13.37 ± 1.43 ^c^	15.67 ± 1.29 ^b^	15.78 ± 1.35 ^b^	17.26 ± 1.54 ^a^	14.17 ± 1.09 ^bc^
Pyridines	12.86 ± 1.06 ^ab^	13.29 ± 1.29 ^ab^	14.11 ± 0.98 ^a^	14.70 ± 1.15 ^a^	14.70 ± 1.43 ^a^	12.57 ± 1.11 ^b^
Phenols	1.21 ± 0.05 ^e^	2.22 ± 0.08 ^d^	2.83 ± 0.11 ^c^	3.11 ± 0.19 ^bc^	3.51 ± 0.23 ^b^	4.96 ± 0.30 ^a^
Pyrrole	14.68 ± 1.05 ^b^	13.66 ± 1.23 ^bc^	13.37 ± 1.17 ^c^	13.37 ± 1.20 ^c^	14.20 ± 0.98 ^bc^	16.82 ± 1.03 ^a^
Ethers	6.66 ± 0.45 ^a^	6.16 ± 0.39 ^a^	3.90 ± 0.21 ^c^	4.28 ± 0.33 ^bc^	3.91 ± 0.29 ^c^	4.54 ± 0.34 ^b^
Total	387.25 ± 25.75 ^b^	402.84 ± 32.88 ^ab^	410.23 ± 27.43 ^ab^	418.05 ± 36.47 ^ab^	449.40 ± 39.66 ^ab^	472.49 ± 40.23 ^a^

Values are expressed as mean ± standard deviation. The superscript letters (a, b, c, d and e) represent statistically significant differences between extraction conditions, as determined through one-way analysis of variance (*p* < 0.05). Control group: conventional cold brew (0.1 MPa, 12 h, 5 °C). The complete table is placed in Appendix A.

**Table 3 foods-12-03857-t003:** Volatile components of UHP-assisted cold brew coffee at different times.

Aroma Type	Content/(μg/L)					
	Time/(Min)					Control Group
	10	15	20	25	30	
Furans	117.34 ± 9.88 ^c^	128.76 ± 8.46 ^bc^	130.93 ± 11.52 ^bc^	134.04 ± 10.24 ^bc^	157.91 ± 12.44 ^a^	142.74 ± 11.29 ^ab^
Aldehydes	81.32 ± 5.35 ^d^	92.59 ± 6.87 ^cd^	94.30 ± 7.55 ^cd^	99.40 ± 8.45 ^bc^	121.43 ± 10.45 ^a^	111.89 ± 9.79 ^ab^
Esters	64.38 ± 4.74 ^c^	71.81 ± 6.52 ^bc^	71.94 ± 5.95 ^bc^	74.43 ± 6.65 ^abc^	79.09 ± 5.41 ^ab^	85.57 ± 6.79 ^a^
Pyrazines	37.77 ± 2.75 ^c^	41.69 ± 3.43 ^bc^	45.63 ± 4.21 ^b^	51.18 ± 3.95 ^a^	53.98 ± 4.02 ^a^	53.34 ± 4.15 ^a^
Alcohols	14.33 ± 1.21 ^d^	16.51 ± 1.43 ^c^	17.52 ± 1.36 ^bc^	19.16 ± 1.55 ^b^	21.81 ± 1.38 ^ab^	25.89 ± 2.02 ^a^
Ketones	9.17 ± 0.62 ^c^	14.20 ± 1.09 ^b^	15.68 ± 1.25 ^ab^	13.50 ± 1.16 ^b^	16.77 ± 1.34 ^a^	14.17 ± 1.29 ^b^
Pyridines	9.12 ± 0.59 ^d^	11.16 ± 0.83 ^c^	14.11 ± 1.26 ^a^	13.21 ± 1.21 ^ab^	13.03 ± 1.09 ^ab^	12.57 ± 0.93 ^b^
Phenols	2.47 ± 0.17 ^d^	2.69 ± 0.15 ^cd^	2.85 ± 0.19 ^c^	2.99 ± 0.16 ^c^	4.27 ± 0.25 ^b^	4.96 ± 0.33 ^a^
Pyrroles	13.27 ± 1.34 ^c^	13.31 ± 0.98 ^c^	13.37 ± 1.25 ^c^	14.00 ± 1.21 ^c^	19.89 ± 1.43 ^a^	16.82 ± 1.39 ^b^
Ethers	1.59 ± 0.12 ^d^	3.46 ± 0.25 ^c^	3.90 ± 0.33 ^b^	4.02 ± 0.38 ^ab^	4.65 ± 0.42 ^a^	4.54 ± 0.39 ^a^
Total	350.77 ± 28.92 ^d^	396.17 ± 33.45 ^cd^	410.23 ± 38.42 ^bcd^	438.07 ± 36.99 ^abc^	507.84 ± 43.68 ^a^	472.49 ± 40.01 ^ab^

Values are expressed as mean ± standard deviation. The superscript letters (a, b, c and d) represent statistically significant differences between extraction conditions, as determined through one-way analysis of variance (*p* < 0.05). Control group: conventional cold brew (0.1 MPa, 12 h, 5 °C). The complete table is placed in Appendix A.

**Table 4 foods-12-03857-t004:** Sensory evaluation of UHP-assisted cold brew coffee under different pressures.

Extraction Conditions		Fruity	Nutty	Sourness	Sweetness	Caramel	Bitterness	Aftertaste	Floral
Pressure/(MPa) 20 min	100	3.25 ± 0.56 ^b^	4.30 ± 0.60 ^c^	4.20 ± 0.42 ^c^	5.20 ± 0.61 ^a^	4.75 ± 0.50 ^d^	6.00 ± 0.66 ^ab^	4.50 ± 0.38 ^c^	2.00 ± 0.50 ^c^
	200	3.88 ± 0.58 ^ab^	5.15 ± 0.51 ^bc^	4.88 ± 0.56 ^bc^	5.00 ± 0.45 ^ab^	5.00 ± 0.47 ^d^	5.88 ± 0.41 ^ab^	5.00 ± 0.65 ^bc^	2.25 ± 0.57 ^bc^
	300	3.88 ± 0.64 ^ab^	5.63 ± 0.60 ^ab^	5.50 ± 0.48 ^ab^	4.70 ± 0.55 ^ab^	5.38 ± 0.47 ^cd^	5.38 ± 0.51 ^b^	5.63 ± 0.60 ^bc^	2.88 ± 0.50 ^abc^
	400	4.25 ± 0.51 ^ab^	5.75 ± 0.62 ^ab^	5.68 ± 0.60 ^ab^	4.00 ± 0.50 ^bc^	6.25 ± 0.64 ^bc^	5.25 ± 0.63 ^b^	5.75 ± 0.59 ^ab^	3.00 ± 0.50 ^abc^
	500	4.50 ± 0.49 ^a^	6.00 ± 0.65 ^ab^	5.85 ± 0.66 ^ab^	3.50 ± 0.52 ^c^	6.63 ± 0.62 ^ab^	5.20 ± 0.48 ^b^	6.13 ± 0.66 ^ab^	3.50 ± 0.48 ^a^
Control group	0.1 MPa 12 h	4.80 ± 0.62 ^a^	6.63 ± 0.36 ^a^	6.50 ± 0.50 ^a^	4.90 ± 0.67 ^ab^	7.50 ± 0.45 ^a^	6.75 ± 0.53 ^a^	6.85 ± 0.57 ^a^	3.25 ± 0.53 ^ab^

Values are expressed as mean ± standard deviation. The superscript letters (a, b, c, and d) represent statistically significant differences between extraction conditions, as determined through one-way analysis of variance (*p* < 0.05). Control group: conventional cold brew (0.1 MPa, 12 h, 5 °C).

**Table 5 foods-12-03857-t005:** Sensory evaluation of UHP-assisted cold brew coffee at different times.

Extraction Conditions		Fruity	Nutty	Sourness	Sweetness	Caramel	Bitterness	Aftertaste	Floral
Holding time/(min) 300 MPa	10	3.13 ± 0.41 ^b^	5.25 ± 0.39 ^b^	5.00 ± 0.52 ^b^	4.00 ± 0.46 ^b^	5.13 ± 0.47 ^c^	4.25 ± 0.68 ^c^	5.20 ± 0.68 ^c^	2.00 ± 0.50 ^c^
	15	3.25 ± 0.62 ^b^	5.63 ± 0.62 ^ab^	5.00 ± 0.49 ^b^	4.30 ± 0.53 ^ab^	5.25 ± 0.52 ^c^	4.85 ± 0.75 ^bc^	5.50 ± 0.50 ^bc^	2.20 ± 0.50 ^bc^
	20	3.88 ± 0.50 ^ab^	5.63 ± 0.58 ^ab^	5.50 ± 0.48 ^ab^	4.70 ± 0.72 ^ab^	5.38 ± 0.81 ^c^	5.38 ± 0.59 ^bc^	5.63 ± 0.66 ^bc^	2.88 ± 0.57 ^abc^
	25	3.88 ± 0.68 ^ab^	6.13 ± 0.38 ^ab^	5.75 ± 0.53 ^ab^	5.13 ± 0.36 ^ab^	5.88 ± 0.45 ^bc^	5.50 ± 0.52 ^abc^	6.13 ± 0.62 ^abc^	3.13 ± 0.26 ^ab^
	30	4.50 ± 0.50 ^a^	6.38 ± 0.60 ^a^	6.25 ± 0.56 ^a^	5.25 ± 0.50 ^a^	6.50 ± 0.50 ^ab^	5.88 ± 0.72 ^ab^	6.50 ± 0.37 ^ab^	3.50 ± 0.50 ^a^
Control group	0.1 MPa 12 h	4.80 ± 0.62 ^a^	6.63 ± 0.36 ^a^	6.50 ± 0.50 ^a^	4.90 ± 0.67 ^ab^	7.50 ± 0.45 ^a^	6.75 ± 0.53 ^a^	6.85 ± 0.57 ^a^	3.25 ± 0.53 ^a^

Values are expressed as mean ± standard deviation. The superscript letters (a, b, and c) represent statistically significant differences between extraction conditions, as determined through one-way analysis of variance (*p* < 0.05). Control group: conventional cold brew (0.1 MPa, 12 h, 5 °C).

## Data Availability

The data that support the findings of this study are available from the corresponding authors upon reasonable request.

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
