# Peer review of "Evaluation of Physicochemical Characteristics and Sensory Properties of Cold Brew Coffees Prepared Using Ultrahigh Pressure under Different Extraction Conditions"

_foods, 2023, doi:10.3390/foods12203857_

Round 1

Reviewer 1 Report

Interesting, important and well-designed, executed (for the most part), analyzed and presented research on the use of high-pressure brewing of cold brew.

All sections are appropriately written, and all tables and figures appear relevant.

The English grammar could use one more proof read.

The sensory analysis section in the materials and methods must be re-written as it is somewhat confusing. The SCA cupping protocol is typically used for the evaluation of the coffee's quality on a 100-point scale. What the researchers did looks more like a descriptive analysis (the rating of the intensity of select attributes). And cupping is not ususally done in clear glasses... Please provide a detailed description of the actual evaluation protocols, the training of the judges, how many replicates, measures of their performance (how well did they discriminate among the coffees, were they reproducible, assuming there were replicate tastings of the coffees, and how good was their alignment - did they rate the coffees the same way? Analysis of variance of their ratings would provide that information). A panel of 6 judges is very small...

Please rephrase "... softer taste" in the abstract as it was not one of the attributes evaluated?

Include the cold brew research of the Ohio State (i.e., Peterson, Simmons) and UC Davis (i.e., Batali et al., Foods, 2022) groups, as they detail differences in sensory attributes between cold brew and hot brew, in your introduction (and possibly discussion).

Reasonably good, but please proof read one more time.

Reviewer 2 Report

This study investigates for a more efficient method for shortening the extraction time of cold brew coffee.The manuscript begins with a well-structured introduction that effectively sets the stage for the study. Experiments have been designed and conducted in detail.

The results should be reviewed according to the choice of data to be included in the tables. I believe this action is necessary to improve the manuscript, make changes to the tables by making a choice aimed at enhancing the results of your research.

The manuscript presents a serious criticality represented by the tables, it is very complicated to read a manuscript that presents 10 pages of tables!

I will only have minor comments for some parts:

1)     What was used to cleanse the palate and minimize the carry-over effect?

2)     Considering that these are “nerve drinks” and was informed consent obtained from the participants?

3)     Did the samples have significantly different sensory attribute scores compared to the control? You can indicate in the caption which attributes are significantly different in according post hoc analysis.

4)     This statement “UHP method has  the advantages of a short extraction time, high extraction rate, and low energy consumption “requires a bibliographic reference

Tables 2-3: It would be appropriate to improve the quality of the tables, they are too long. I ask the authors for a better graphical representation. I advise the authors to use the table data to implement the sensory data, the remaining data entered in the tables could be placed in the supplementary material.

Reviewer 3 Report

The manuscript "Evaluation of physicochemical characteristics and sensory properties of cold brew coffees by using ultra-high pressure under different extraction conditions” by Shiyu Chen and colleagues is an interesting research work. In general, the methodology and experimental work are correct and understanding to the readers.

However, in my opinion, authors could improve the manuscript introducing a few changes in several specific points:

1 - All determinations were done in duplicate/triplicate ? this is not clear in material and methods. Authors, in line 181 state that the tests were carried out in triplicate, but the number of repetitions carried out for each parameter analyzed is not clear.

2 - Result section and Discuss section could be combined in a single section: Results and discussion. In fact, in the discussion section, authors are repeating a description of the results. The authors are already discussing the results, considering the results of other works and possible explanations. Therefore, it would also be good to add an item of general conclusions of the work.

3 - Line 265-266: “minor volatile compounds” what it means ? is related to low concentrations ? Clarify it.

4 - Line 289: So, in that case, this was confirmed by the results of the sensory evaluation ?.

5 - Line 302-324: The authors could introduce in a clearer and more evident way whether the results of the evaluation of volatile compounds were confirmed or not by the sensory evaluation. Therefore, it would be good to introduce a better relationship between these two “types” of results. On the other hand, could be positive to introduce in tables 2 and 3 a new column containing the sensory descriptor (when it exists) for each volatile compound quantified and the respective reference.

6 - For the results in radar map (figure 2). Could be very interesting also to introduce a statistical work to compare the average results for each sensory parameter between each of the samples. For example, an ANOVA analysis.
